# How Can Imaging Help the Radiation Oncologist in Multiple Myeloma Treatment

**DOI:** 10.3390/medicina57010020

**Published:** 2020-12-28

**Authors:** Liliana Belgioia, Stefano Vagge, Alberto Tagliafico, Renzo Corvò

**Affiliations:** 1Health Science Department (DISSAL), University of Genoa, 16132 Genoa, Italy; alberto.tagliafico@unige.it (A.T.); renzo.corvo@unige.it (R.C.); 2Radiation Oncology Department, IRCCS Ospedale Policlinico San Martino, 16132 Genoa, Italy; stefano.vagge@hsanmartino.it; 3Department of Radiology, IRCCS Ospedale Policlinico San Martino, 16132 Genoa, Italy

**Keywords:** myeloma multiple, radiation therapy, imaging, SBRT, antalgic effect

## Abstract

Multiple myeloma is an incurable malignant tumor of plasma cells of the bone marrow; most patients present a disseminated disease with important bone involvement. Even though a chemotherapy-based approach is the major treatment, radiotherapy often has a supportive role for symptom relief but also a radical role for patients with indolent disease or localized forms. In both cases imaging is the basis for treatment planning and for correct patient classification. This paper aims to describe and summarize how radiation oncologists could use imaging information to personalize the treatment for each patient.

## 1. Introduction

Multiple myeloma (MM) is a malignant neoplasm of plasma cells that accumulate in the bone marrow leading to bone destruction and marrow failure. Its incidence in Europe is about 1% and represents over 10% of hematologic disease [1]. Although treatment of MM is mainly chemotherapy-based, radiotherapy (RT) also could be very useful for patients even if, in the majority of cases, it has a palliative role [2]. It is well known that in MM there is a complex interaction between MM cells, bone marrow stromal cells and osteoclasts that lead to an increased bone reabsorption by osteoclasts and reduced bone formation through suppression of osteoblast function [3]. The direct consequence of this mechanism is the appearance of osteolytic bone lesions with possible complications and the onset of symptoms that range from pain, pathologic fractures to spinal compression [3]. Moreover, considering that in the last decades, systemic therapies have evolved with subsequent improvement of oncological outcomes and longer survival, RT could contribute to obtaining local control and symptom relief for a large group of patients [4]. The aim of this paper is to highlight the role of radiotherapy in multiple myeloma with particular attention to imaging and modern radiotherapy techniques.

## 2. Palliative/Consolidation Radiotherapy

Osteolytic lesions are present at diagnosis in about 70–80% of patients, larger osteolytic lesions are frequently associated with pain and fracture risk. Pathological fractures could occur in 40% of patients in the first year after diagnosis and 60% of patients during the disease course [5]. The use of radiotherapy for the treatment of bone lesions could help reduce or prevent these skeletal-related events and maintain or improve patients’ quality of life [2]. In this setting of patients, the aims of RT are mainly two: obtain pain control and recalcification. Antalgic effect could be reached in about 75–90% of patients, obtained during or immediately after RT and could last for a long period [2,6]. The mechanisms of analgesic effect from irradiation are not completely understood but it is mainly due to the inhibition of pain mediators and to the shrinkage of the tumor. Conversely, recalcification is a long-term effect that appears after months. Matuscheck et al. [7], in a retrospective analysis of 69 patients, evaluated 108 irradiated lesions for changes in calcification based on pre- and post-treatment radiographs. The overall recalcification rate was 48% (23 and 25% of lesions obtained full and partial recalcification, respectively). No change of calcification could be documented in 42% of the bone lesions, and 10% of the patients developed progression of osteolytic lesions. The authors detected that higher total RT doses were significantly associated with better recalcification [7] and indicated an increase from 20 to 30 Gy in total dose correlates with a 12% higher likelihood of recalcification. This correlation is confirmed by several other data available in literature, several studies reported that recalcification was achieved in 40–50% of the irradiated bone destructions [8,9]. The possibility of obtaining recalcification is obviously extremely important as it reduces the risk of fracture or bone complications. These data are derived in the majority of studies from evaluation of conventional skeletal radiography but are also confirmed by evaluation of CT imaging.

Low-dose whole-body CT is recommended over conventional skeletal surveys to help diagnose and stage multiple myeloma bone disease; this last method can be used as well as whole-body CT or when other novel imaging methods are not available [10]. To date, a diagnostic CT scan is mandatory for better identification and delineation of a target for RT treatment, especially for lesions localized in the spine and pelvis more than in long bones [11]. Moreover, CT imaging is the first choice to identify and assess the extension of lytic lesions and evaluate possible extraosseous extension in soft tissue. The decision to treat a patient with RT should also consider the possible risk of pathological fractures or neurological complications. However, the high-risk lesions should be first stabilized by orthopedic measures and combined with postoperative radiation treatment to improve pain and local control. 

The CT scan helps identify an impending fracture, for which definition varies according to the major anatomical sites (long bones, vertebra or acetabulum), and detect important factors such as cortical bone destruction, location and extension of lesion, vertebral body collapse of >50% or transitional deformity that can lead to radiation treatment [12]. In literature, several scores are described to identify bone metastases at high risk of injury and the majority are based on radiographic imaging and clinical factors [13,14]. Recently, a simple scoring method specific for MM has been introduced; this method, called the Myeloma Spine and Bone Damage Score (MSBDS) was developed with the final aim to provide a semiquantitative tool to evaluate the status of bone damage and risk of fracture and instability on a standard total-body CT used in the routine practice of MM centers. The total score range from 0 to >10, where a value > 10 identifies high risk lesions that need immediate surgical or radiation oncologist consultation, a score ≥ 5–10 medium risk lesions for possible instability or pathologic fracture and a score < 5 low risk lesions [15]. 

In the case of a suspected neurological complication, the best radiological method is MRI especially if the involvement of soft tissue is present [16]. MRI helps discriminate myeloma from normal marrow and differentiate between myeloma and benign causes of vertebral fracture [17]. MRI is of extreme importance in the cases of patients with vertebral fracture and no other signs of disease or lytic lesions, to evaluate a painful lesion, mainly in the axial skeleton, or preferably in the evaluation of collapsed vertebrae, especially when myeloma is not active, where the possibility of osteoporotic fracture is high [16]. Moreover, MRI is very useful for detecting spinal cord compression: this severe clinical condition can occur in 10% to 20% of patients with MM and the most commonly involved site is the dorsal spine, followed by the lumbar and sacral regions [18]. In this last clinical scenario, RT has an established role both after decompressive surgery and on its own; we have to keep in mind that MM presents a high radiosensitivity, so good results are also obtained in these emergencies with RT alone. 

Rades et al. [19] retrospectively analyzed the data of 237 patients with motor deficits of the lower extremities for spinal cord compression from vertebral body myeloma that did not undergo upfront surgery but were treated with RT alone. The authors reported an excellent response rate (97%), with 88% of the patients walking after RT and 64% that regained the ability to walk. The overall local control rates were 93, 82, and 82% at 1, 2, and 3 years, respectively [19]. 

Regarding radiation dose, it is suggested that for epidural disease with spinal cord compression, a higher total dose should be prescribed when durable local control is desired compared to radiotherapy with an antalgic role. The International Lymphoma Radiation Oncology Group guidelines suggest a hypofractionated regimen with a total dose of 8 to 30 Gy for symptom relief, with a recommendation to use a single fraction of 8 Gy in patients with poor prognosis but a dose of 30 Gy in 10 fraction in case of spinal cord involvement [4]. 

## 3. Stereotactic Body Radiotherapy (SBRT)

Stereotactic body radiotherapy (SBRT) is a technique that allows the delivery of an ablative radiation dose in a few fractions up to a single fraction in stereotactic radiosurgery (SRS), with concomitant sparing of normal tissues [20]. SBRT is not yet a standard treatment in MM, but in selected patients might be a promising opportunity that should be investigated in clinical trials. To date, few data on SBRT/SRS in MM are available in the literature. Jin R et al. reported the outcomes of 24 patients with epidural spinal cord compression treated with SRS on 31 lesions. Median single fraction radiation dose was 16 Gy (range: 10−18 Gy), administered to the involved spine including the epidural or paraspinal tumor. With a median follow up of 11.2 months, overall pain control rate was 86%; complete and partial relief were obtained in 54 and 32% of the patients, respectively. Complete radiographic response of the epidural tumor was noted in 81% at 3 months after radiosurgery. Authors reported extremity weakness and sensory deficits in 7 patients, of which 5 made complete neurological recoveries in 1−6 months [21]. 

Miller et al. instead, reported a series of 56 lesions in 38 MM patients treated with spine SRS; the primary endpoint was pain relief and secondary endpoints were incidences of radiographic failure and vertebral fracture. Epidural disease, pre-existing vertebral fracture, thecal sac compression and neural foraminal involvement were present in 77, 63, 55 and 48% of treatment sites, respectively. Moreover, 30% of treated sites were previously irradiated. The median prescription dose was 14 Gy (range 10–24 Gy) in a single fraction (range: 1–4). Vertebral fracture occurred following 12 treatments (21%), with an 18% cumulative incidence of fracture at 6 and 12 months. Two patients (4%) developed pain flare following spine SRS. A rapid and durable symptomatic response was observed, with a median time to pain relief of 1.6 months. This response was durable among 85% of patients at 12 months following treatment, with 91% local control [22]. The critical issue of SBRT/SRS is the risk of fracture that may be related to the collapse of bone into the tumor cavity after ablation. The literature data on patients with solid tumors reported a rate of vertebral fracture after SBRT, closer to 14% [23,24] compared to 3% for conventional radiotherapy [25]; in particular, patients with lytic tumors, spinal misalignment and baseline fracture seemed at a higher risk to develop a vertebral fracture. The other critical point of this kind of treatment is that safe delivery of RT is imperative to avoid irreversible neurological sequelae; from this point of view, the advances in treatment planning, immobilization, and treatment delivery have been relevant. If a spinal SBRT is proposed, MRI is highly recommended. After the acquisition of a planned CT scan, T1 and T2 weighted MRI sequences are fused to delineate target and critical structures [26]. The guidelines for definition of target volumes in patients receiving upfront SRS were published in 2012 by International Spine Radiosurgery Consortium Consensus [26] and updated for the postoperative setting in 2017 [27]. These guidelines are not specific for MM disease but are general indications for target delineation of all malignant metastases to the spine. Patient immobilization, and a pretreatment image-guided radiotherapy (IGRT) like a cone-beam CT or a MVCT as pretreatment verification, are mandatory to allow for interfraction reproducibility, minimize planning target volumes, sculpt dose to intended targets, and avoid neurologic toxicities. We underline that SBRT/SRS in patients with MM is not yet a standard of care and might be investigated in the context of a clinical trial, especially in highly selected situations such as re-irradiation. Response evaluation after spinal SBRT/SRS is a dark spot because RECIST criteria are not easy to apply. A first report from the SPIne response assessment in Neuro-Oncology (SPINO) group was published in 2015 and recognized MRI as the main method for the optimal response assessment [28]. This report takes into consideration the so-called pseudoprogression (PP), defined as a treatment-related tumor growth that simulates progression of disease [29]. This phenomenon is similar to that described for brain radiosurgery, lung or liver SBRT, but data on osseous PP are limited. The first report in literature on PP was published by Al Omair et al. in 2013 for 2 patients that presented increased pain at 2 and 3 years after SBRT; the treated sites appeared larger at MRI but the subsequent biopsy revealed necrotic or inflammatory tissue [30]. PP is not rare with reported incidence of 14%–18% in different series and, in general, with an early detection after a few months from SRS (3.5–5 months) [31,32]. The most common findings at MRI are increased T2 signal intensity combined with dark signal intensity changes on both morphologic and fluid-sensitive sequences and stable or decreased tumor volume [33]. 

## 4. Radical Radiotherapy

Radiotherapy has a radical role in solitary plasmacytoma (SP), as it is a localized accumulation of neoplastic plasma cell without systemic involvement [4]. According to the location, SPs could be divided into solitary bone plasmacytoma (SPb) and solitary extramedullary plasmacytoma (SPe) [34]. The latter is less common (about 20–30% of SP diagnosis) and can occur in any organ and tissue but about 80% arise in the head and neck region. In this situation, RT is the mainstay of treatment; in fact, patients who received RT have a lower rate of local relapse than those who did not receive it, with an excellent local control rate of 79–91% that translates into long remission and even cure [35].

To date the optimal dose of RT is still not established, but some retrospective data have indicated to tailor the dose according to the size of the disease and ILROG guidelines suggest a total dose of 35–40 Gy in case of SPb extending less than 5 cm and 40 to 50 Gy for SPb > 5 cm or SPe [4].

In these sets of patients, it is essential to start with the correct assessment of the disease staging, both local and systemic, because it can radically change the therapeutic approach. CT scan remains the basis for RT planning, but it is strongly suggested before RT to perform MRI and 18 FDG PET/CT [36]. 

MRI is indispensable for head and neck lesions to investigate the local spread of disease and its possible infiltration of the nearest structures; it can help with target delineation both of primary tumor and regional nodes. SPe diagnosed in the head and neck region can present an involvement of 25% of regional nodes [37], and if positive nodes are detected they have to be included in CTV, while prophylactic node irradiation seems to be not useful unless there is clinical evidence of a high risk nodal involvement such as bulky disease, according to ILROG panel consensus [38]. 

Another imaging method that is mandatory in SP is 18 FDG PET/CT (International Myeloma working group guidelines), as it is highly sensitive for detecting myeloma deposits. Some data show that it can reveal additional lesions in about 30% of patients compared to MRI [39]. For radiation oncologists, it can be crucial at baseline to detect disease at a nodal level to impact target delineation; moreover, it is essential in the evaluation of the response to treatment and for the follow up of patients across years. With regard to the first point, PET/CT is able to differentiate between metabolically active and nonactive disease, some data suggest that PET-CT demonstrated a prompt normalization of imaging findings in patients with good or complete clinical response to therapy that is faster than those of MRI, so it is very useful for patients in which post-treatment MRI remains positive [40]. 

To conclude in SP from a radiation point of view, MRI and PET/CT offer complementary information and guarantee a precise evaluation of the disease; they can be helpful first in the definition of GTV if fusion with planning CT is feasible and also in the follow up of patients.

Another setting in which radiotherapy continues to have an important role is in patients undergoing hematopoietic cell transplantation; in fact, RT in the form of total body irradiation, continues to be an important part of the conditioning regimen with chemotherapy. Technological innovation has allowed more precise radiation therapy delivery and a dose escalation in the sites considered to be at a higher risk. In MM this is translated in irradiation of all skeletal bones with a technique called total marrow irradiation. In this setting imaging is of particular importance; especially PET/CT could guide the radiation oncologist to identify the target to boost with increased doses before hematopoietic cell transplantation or in patients with relapsed or progressive disease [41]. 

## 5. Conclusions

Radiotherapy is crucial in the management of patients with multiple myeloma in a palliative and in a radical setting. Information from the different imaging modalities help the radiation oncologist to correctly stage the patient, to put indications for radiation treatment, to identify target volumes and to evaluate the response to treatment. Different imaging modalities widely used today such as MRI and CT give the radiation oncologist complementary information to tailor RT. 

## Data Availability

No new data were created or analyzed in this study. Data sharing is not applicable to this article.

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
