# Peer review of "How Can Imaging Help the Radiation Oncologist in Multiple Myeloma Treatment"

_medicina, 2020, doi:10.3390/medicina57010020_

Round 1

Reviewer 1 Report

Original work. Very well structured, with correct methodology. This work can make an important contribution to the subject matter. No substantial relief or correction to be made.

Author Response

Thank you for your positive comments.

Reviewer 2 Report

In this paper, Dr. Belgioia and colleagues review the role of radiologic imaging in the radiation oncology treatment of localized and disseminated plasma cell disorders.

The paper is of value and clearly describes the utility of different radiologic exams in guiding radiation oncologists in the planning phase.

I have some comments:

  • The authors cite a previous paper, demonstrating better recalcification with higher RT doses: it could be of value for the readers to get more details on the doses of radiation required to achieve a valuable recalcification.

  • The authors should detail the doses of radiation recommended for the treatment of solitary plasmacytoma

Minor Comments:

The manuscript has several grammar mistakes and could benefit from a detailed language editing.

Line 15: I suggest to replace “for personalized” with “to personalize”

Line 21: I suggest to remove “of” before 1%

Line 29: I suggest to modify “systemic therapy” with “systemic therapies”

Line 40: replace “could lasted” with “could last” 

Line 51: replace “it reduce” with “it reduces”

Line 51: between “data” and “derived” is missing “are”

Line 100: replace “allow” with allows”

Line 125: replace “seem a higher risk” with “seem at higher risk”

Line 135: replace “an MVCT” with “a MVCT”

Line 139: replace “Evaluates response” with “Response evaluation”

Line 187: the abbreviation TMI is unnecessary.

Author Response

Thank you for your comments. We changed the text according to your observations.  We reviewed English language.

Reviewer 3 Report

The manuscript reports the use of imaging, i.e. CT, MRI, PET/CT to detect MM lesions and for radiotherapy planning.

All of this is well known and used in clinical routine. The manuscript does not add anything really new to the knowledge in the field.

The more interesting part of the manuscript is about stereotactic radiotherapy of MM lesions, which is actually not about imaging (of course adequate imaging is always relevant for stereotactic radiotherapy). Even in this part, the manuscript does not add new knowledge.

Author Response

Thank you for your observations. We tried to give an overview on actual data on imaging in multiple myeloma  from a radiation oncologist' point of view  from palliative to radical treatment. To improve the manuscript we reviewed English language.

Round 2

Reviewer 3 Report

The work is not a systematic review and could be improved by revealing the search algorithm for the evaluated literature etc.

Author Response

Thank you for your suggestion. Computerized search from PubMed was conducted by a independent researcher with a specialization in radiation oncology. The selection of keywords and related Medical Subject Headings (MeSH terms) were the following: “multiple myeloma”, “solitary plamacytoma”, “radiotherapy”, “SBRT”, “Total Marrow Irradiation”, “imaging”, “MRI”, CT”, “PET/CT”. The search was restricted to English article. It was added in the text.